# PeerJ

# PeptideBuilder: A simple Python library to generate model peptides

Matthew Z. Tien[1], Dariya K. Sydykova[2], Austin G. Meyer[2,3] and Claus O. Wilke[2]

[1] Department of Biochemistry & Molecular Biology, The University of Chicago, Chicago, IL, USA
[2] Section of Integrative Biology, Institute for Cellular and Molecular Biology, and Center for Computational Biology and Bioinformatics, The University of Texas at Austin, Austin, TX, USA
[3] School of Medicine, Texas Tech University Health Sciences Center, Lubbock, TX, USA

## ABSTRACT

We present a simple Python library to construct models of polypeptides from scratch. The intended use case is the generation of peptide models with pre-specified backbone angles. For example, using our library, one can generate a model of a set of amino acids in a specific conformation using just a few lines of python code. We do not provide any tools for energy minimization or rotamer packing, since powerful tools are available for these purposes. Instead, we provide a simple Python interface that enables one to add residues to a peptide chain in any desired conformation. Bond angles and bond lengths can be manipulated if so desired, and reasonable values are used by default.

## INTRODUCTION

Researchers working in structural biology and related fields frequently have to create, manipulate, or analyze protein crystal structures. To aid this work, many different software tools have been developed. Examples include visualization (*Schrödinger, 2013*), mutagenesis (*Schrödinger, 2013*; *Leaver-Fay et al., 2011*), high-throughput computational analysis (*Hamelryck & Manderick, 2003*; *Grant et al., 2006*), ab-initio protein folding and protein design (*Leaver-Fay et al., 2011*), and homology modeling and threading (*Eswar et al., 2006*; *Zhang, 2008*). In comparison, a relatively simple task, the ab-initio creation of a protein structure in a desired conformation, has received little attention. It is possible to perform this task in PyRosetta (*Chaudhury, Lyskov & Gray, 2010*; *Gray et al., 2013*), but that approach incurs the overhead of the entire Rosetta protein modeling package (*Leaver-Fay et al., 2011*). One can also construct peptides manually in some graphical molecular modeling packages, such as Swiss-PdbViewer (*Guex & Peitsch, 1997*). Finally, the Rose lab has developed Ribosome (*Srinivasan, 2013*), a small program with the express purpose of creating model peptides. However, Ribosome is implemented in Fortran, an outdated programming language that integrates poorly with modern bioinformatics pipelines.

For a recent analysis by our group, we wanted to systematically enumerate GLY-X-GLY tripeptides in all allowed conformations (*Tien et al., 2012*). After review of the available

Corresponding author
Claus O. Wilke,
wilke@austin.utexas.edu

software packages, we determined that there was a need for a lightweight library, implemented in a modern programming language, that would allow us to construct arbitrary peptides in any desired conformation. We decided to write this library in the language Python (*Python Sofware Foundation, 2013*), as this language is widely used in scientific computing. Specifically, many tools suitable for computational biology and bioinformatics are available (*Cock et al., 2009*), including tools to read, manipulate, and write PDB (Protein Data Bank) files (*Hamelryck & Manderick, 2003*). This effort resulted in the Python library `PeptideBuilder`, which we describe here. The library consists of two Python files comprising a total of approximately 2000 lines of code. Both files are provided as Supplemental Information 1. The entire PeptideBuilder package is also available online at https://github.com/mtien/PeptideBuilder.

## CONCEPTUAL OVERVIEW

The key function our library provides is to add a residue at the C terminus of an existing polypeptide model, using arbitrary backbone angles. Our library also allows a user to generate an individual amino acid residue and place it into an otherwise empty model. In combination, these two functions enable the construction of arbitrary polypeptide chains. The generated models are stored as structure objects using the PDB module of Biopython (`Bio.PDB`, *Hamelryck & Manderick 2003*). The seemless integration with Biopython's PDB module means that we can leverage a wide range of existing functionality, such as writing structures to PDB files or measuring distances between atoms.

Adding a residue to an existing polypeptide chain involves two separate steps. First, we have to establish the desired geometric arrangement of all atoms in the residue to be added. This means we have to determine all bond lengths and angles. In practice, we will usually want to specify the dihedral backbone angles $\phi$ and $\psi$, and possibly the rotamers, whereas all other bond lengths and angles should be set to reasonable defaults for the amino acid under consideration. Once we have determined the desired geometry, we have to calculate the actual position of all atoms in 3-space and then add the atoms to the structure object. The exact calculations required to convert bond lengths and angles into 3D atom coordinates are given in the supporting text of *Tien et al. (2012)*. Our library places all heavy atoms for each residue, but it does not place hydrogens.

We obtained default values for bond lengths and angles by measuring these quantities in a large collection of published crystal structures and recording the average for each quantity, as described (*Tien et al., 2012*). We set the default for the backbone dihedral angles to the extended conformation ($\phi = -120°$, $\psi = 140°$, $\omega = 180°$). We based the default rotamer angles of each individual amino acid on the rotamer library of *Shapovalov & Dunbrack (2011)*. For each amino acid, the rotamer library provided the frequency of each combination of rotamer angles given the backbone conformation. We analyzed this library at the extended backbone conformation ($\phi = -120°$, $\psi = 140°$) and used the most likely rotamer conformation at that backbone conformation as default. For amino acids with multiple $\chi_1$ or $\chi_2$ angles (such as Ile or Leu) we used the most likely and the second most likely angles from the rotamer library.

## EVALUATION

We did extensive testing on our library to verify that we were placing atoms at the correct locations given the bond lengths and angles we specified. First, we collected tri-peptides from published PDB structures, extracted all bond lengths and angles, reconstructed the tri-peptides using our library, and verified that the original tri-peptide and the reconstructed one aligned with an RMSD of zero. Next, we wanted to know how our library would fare in reconstructing longer peptides, in particular when using the default parameter values we used for bond lengths and angles. For this analysis, we focused on the peptide backbone, since the evaluation of tripeptides had shown that our library was capable of placing side-chains correctly if it was given the correct bond lengths and angles.

We selected ten proteins with solved crystal structure. The proteins were chosen to represent a diverse group of common folds. For each protein, we then attempted to reconstruct the backbone of either the first 50 residues in the structure, the first 150 residues in the structure, or the entire structure. In all cases, we extracted backbone bond lengths and angles at each residue, and then reconstructed the protein using four different methods. When placing each residue, we either (i) adjusted only the extracted $\phi$ and $\psi$ dihedral angles, (ii) adjusted $\phi$, $\psi$, and $\omega$ dihedral angles, (iii) adjusted all dihedral and planar bond angles, or (iv) adjusted all bond lengths and angles exactly to the values measured in the structure we were reconstructing. In each case, any remaining parameters were left at their default values.

As expected, when we set all bond lengths and angles to exactly the values observed in the reference crystal structure, we could reconstruct the entire backbone with an RMSD close to zero. We did see an accumulation of rounding errors in longer proteins, but these rounding errors amounted to an RMSD of less than 0.01 Å even for a protein of over 600 residues. Hence they are negligible in practice. By contrast, reconstructions relying on just backbone dihedral angles performed poorly. We found that we had to adjust all backbone bond angles, inlcuding planar angles, to obtain accurate reconstructions. Bond lengths, on the other hand, could be left at their default values. Table 1 summarizes our findings for all 10 structures, and Fig. 1 shows the results of the four different methods of reconstruction for one example structure. The python script to generate these reconstructions is provided as part of Supplemental Information 1.

Our results show that the `PeptideBuilder` software correctly places all atoms at the desired locations. However, they also demonstrate that one needs to be careful when constructing longer peptides. It is not possible to construct an entire protein structure just from backbone dihedral angles and expect the structure to look approximately correct. In particular in tight turns and unstructured loops, small deviations in backbone bond angles can have a major impact on where in 3D space downstream secondary structure elements are located. Hence these angles cannot be neglected when reconstructing backbones.

## USAGE

The `PeptideBuilder` software consists of two libraries, `Geometry` and `PeptideBuilder`. The `Geometry` library contains functions that can set up the proper

**Peer**J

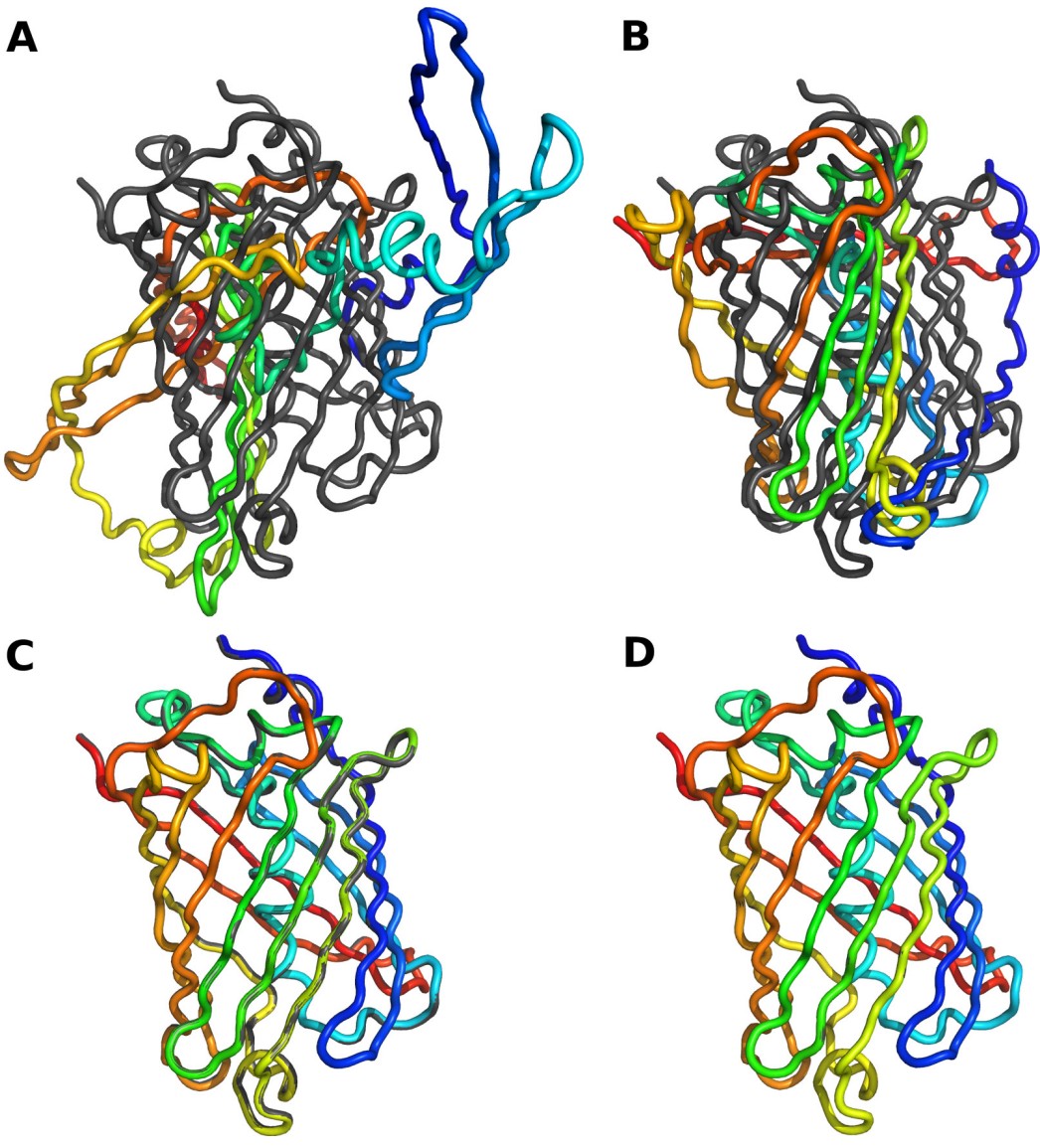

**Figure 1 Reconstruction of protein backbone using varying degrees of modeling accuracy.** The gray backbone corresponds to chain A of crystal structure 1gfl (green fluorescent protein), and the rainbow-colored backbone corresponds to the reconstructed version thereof. (A) Only $\phi$ and $\psi$ dihedral angles are adjusted to match those in the reference structure. (B) All dihedral backbone angles ($\phi$, $\psi$, and $\omega$) are adjusted to match those in the reference structure. (C) All backbone bond angles are adjusted to match those in the reference structure. (D) All backbone bond lengths and angles are adjusted to match those in the reference structure. RMSD values are given in Table 1. Part (D) shows perfect overlap between the reference and the reconstructed backbone.

three-dimensional geometry of all 20 amino acids. The `PeptideBuilder` library contains functions that use this geometry information to construct actual peptides.

`PeptideBuilder` has one dependency beyond a default python installation, the Biopython package (*Cock et al. 2009*, http://biopython.org/), which provides the module `Bio`.

**Table 1** Root mean square deviation (RMSD) between reference crystal structures and reconstructions of these structures using varying amounts of modeling detail. Reconstructions using just dihedral backbone angles tend to deviate substantially from the reference structures, whereas reconstructions using all bond angles tend to perform well, even if bond lengths are kept at default values.

| PDB[a] | Length[b] | $\phi, \psi$ [c] | | | $\phi, \psi, \omega$ [d] | | | All bond angles | | | All bond lengths and angles | | |
|---|---|---|---|---|---|---|---|---|---|---|---|---|---|
| | | 50[e] | 150[f] | full[g] | 50[e] | 150[f] | full[g] | 50[e] | 150[f] | full[g] | 50[e] | 150[f] | full[g] |
| 1aq7 | 223 | 5.8 | 10.4 | 11.6 | 5.0 | 8.9 | 10.1 | 0.0 | 0.1 | 0.1 | 0.0 | 0.0 | 0.0 |
| 1gfl | 230 | 3.9 | 21.3 | 22.4 | 4.0 | 7.4 | 9.0 | 0.1 | 0.1 | 0.1 | 0.0 | 0.0 | 0.0 |
| 1nbw | 606 | 4.9 | 8.5 | 29.1 | 4.3 | 7.4 | 25.6 | 0.0 | 0.1 | 0.2 | 0.0 | 0.0 | 0.0 |
| 1vca | 199 | 6.2 | 20.1 | 23.9 | 2.8 | 8.7 | 11.7 | 0.1 | 0.2 | 0.2 | 0.0 | 0.0 | 0.0 |
| 2o6r | 177 | 3.6 | 12.7 | 14.9 | 3.0 | 10.4 | 12.1 | 0.1 | 0.1 | 0.1 | 0.0 | 0.0 | 0.0 |
| 2r83 | 279 | 7.1 | 14.4 | 17.1 | 5.6 | 11.7 | 15.9 | 0.0 | 0.1 | 0.1 | 0.0 | 0.0 | 0.0 |
| 3cap | 326 | 3.9 | 7.6 | 10.6 | 2.0 | 6.1 | 9.4 | 0.0 | 0.1 | 0.1 | 0.0 | 0.0 | 0.0 |
| 3cuq | 219 | 5.2 | 9.9 | 9.3 | 3.2 | 5.2 | 5.2 | 0.1 | 0.1 | 0.2 | 0.0 | 0.0 | 0.0 |
| 3vni | 289 | 5.0 | 12.7 | 16.7 | 3.2 | 6.6 | 7.1 | 0.1 | 0.2 | 0.2 | 0.0 | 0.0 | 0.0 |
| 7tim | 247 | 5.3 | 6.7 | 10.3 | 4.8 | 6.2 | 7.6 | 0.1 | 0.2 | 0.3 | 0.0 | 0.0 | 0.0 |

**Notes.**

[a] PDB ID of the reference structure. In all cases, chain A of the structure was used.

[b] Length of reference structure, in amino acids.

[c] Only dihedral backbone angles $\phi$ and $\psi$ have been adjusted to match the reference structure.

[d] Only dihedral backbone angles $\phi$, $\psi$, and $\omega$ have been adjusted to match the reference structure.

[e] RMSD (in Å) over the first 50 residues.

[f] RMSD (in Å) over the first 150 residues.

[g] RMSD (in Å) over the entire structure.

## Basic construction of peptides

Let us consider a simple program that places 5 glycines into an extended conformation. First, we need to generate the glycine geometry, which we do using the function `Geometry.geometry()`. This function takes as argument a single character indicating the desired amino acid. Using the resulting geometry, we can then intialize a structure object with this amino acid, using the function `PeptideBuilder.initialize_res()`. The complete code to perform these functions looks like this:

```
1 import Geometry, PeptideBuilder
2 geo = Geometry.geometry("G")
3 structure = PeptideBuilder.initialize_res(geo)
```

We now add four more glycines using the function `PeptideBuilder.add_residue()`. The default geometry object specifies an extended conformation ($\phi = -120°$, $\psi = 140°$). If this is the conformation we want to generate, we can simply reuse the previously generated geometry object and write:

```
4 for i in range(4):
5         structure = PeptideBuilder.add_residue(structure, geo)
```

Because `PeptideBuilder` stores the generated peptides in the format of `Bio.PDB` structure objects, we can use existing `Bio.PDB` functionality to write the resulting structure into a PDB file:

**Table 2  Overview of functions provided by PeptideBuilder.**

| Function name | Description |
| --- | --- |
| add_residue | Adds a single residue to a structure. |
| initialize_res | Creates a new structure containing a single amino acid. |
| make_structure | Builds an entire peptide with specified amino acid sequence and backbone angles. |
| make_extended_structure | Builds an entire peptide in the extended conformation. |
| make_structure_from_geos | Builds an entire peptide from a list of geometry objects. |

```
6  import Bio.PDB # import Biopython's PDB module.
7  out = Bio.PDB.PDBIO()
8  out.set_structure(structure)
9  out.save( "example.pdb" )
```

If we want to generate a peptide that is not in the extended conformation, we have to adjust the backbone dihedral angles accordingly. For example, we could place the five glycines into an alpha helix by setting $\phi = -60°$ and $\psi = -40°$. We do this by manipulating the phi and psi_im1 members of the geometry object. (We are not actually specifying the $\psi$ angle of the residue to be added, but the corresponding angle of the previous residue, $\psi_{i-1}$. Hence the member name psi_im1. See the Detailed adjustment of residue geometry Section for details.) The code example looks as follows:

```
1  geo = Geometry.geometry("G")
2  geo.phi=-60
3  geo.psi_im1=-40
4  structure = PeptideBuilder.initialize_res(geo)
5  for i in range(4):
6          structure = PeptideBuilder.add_residue(structure, geo)
```

Several convenience functions exist that simplify common tasks. For example, if we simply want to add a residue at specific backbone angles, we can use an overloaded version of the function add_residue() that takes as arguments the structure to which the residue should be added, the amino acid in single-character code, and the $\phi$ and $\psi_{i-1}$ angles:

```
1  # add an arginine, setting phi=-60 and psi_im1=-40
2  structure = PeptideBuilder.add_residue(structure, "R", -60, -40)
```

If we want to place an arbitrary sequence of amino acids into an extended structure, we can use the function make_extended_structure(), which takes as its sole argument a string holding the desired amino-acid sequence:

```
1  # construct a peptide corresponding to the
2  # sequence "MGGLTR" in extended conformation
3  structure = PeptideBuilder.make_extended_structure("MGGLTR")
```

Table 2 summarizes all functions provided by PeptideBuilder. All these functions are documented in the source code using standard Python self-documentation methods.

## Detailed adjustment of residue geometry

Geometry objects contain all the bond lengths, bond angles, and dihedral angles necessary to specify a given amino acid. These parameters are stored as member variables, and can be

**Peer**J

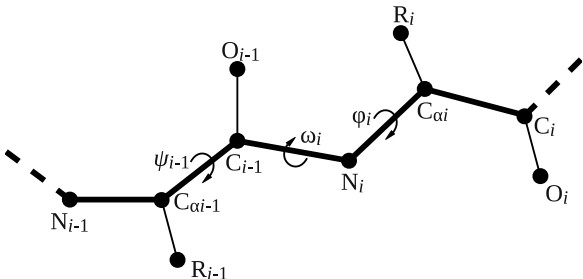

**Figure 2 Illustration of backbone dihedral angles.** When placing the atoms for residue *i*, we have to specify the $\phi$ and $\omega$ dihedral angles for that residue ($\phi_i$ and $\omega_i$) and the $\psi$ angle for the previous residue ($\psi_{i-1}$). The $\psi$ angle for residue *i* involves the nitrogen atom of residue $i+1$ and thus remains undefined until residue $i+1$ is added.

changed by assignment (as in `geo.phi=-60`). We use a uniform naming scheme across all amino acids. Member variables storing bond lengths end in `_length`, those storing bond angles end in `_angle`, and those storing dihedral angles end in `_diangle`. For example, `CA_N_length` specifies the bond length between the $\alpha$ carbon and the backbone nitrogen, and `CA_C_O_angle` specifies the planar bond angle between the $\alpha$ carbon, the carbonyl carbon, and the carbonyl oxygen. All bond lengths are specified in units of Å, and all angles are specified in degrees.

Four backbone geometry parameters deviate from the uniform naming scheme: the three backbone dihedral angles $\phi$ (`phi`), $\psi_{i-1}$ (`psi_im1`), and $\omega$ (`omega`) and the length of the peptide bond (`peptide_bond`). Figure 2 visualizes the location of the backbone dihedral angles in the peptide chain. The parameter `phi` places the carbonyl carbon of the residue to be added relative to the previous amino acid in the peptide chain, so it corresponds to the $\phi$ angle of the residue being placed ($\phi_i$). The parameter `psi_im1` places the nitrogen atom of the residue to be added, and hence corresponds to the $\psi$ angle of the preceding residue in the chain ($\psi_{i-1}$). The parameter `omega` places the $\alpha$ carbon relative to the preceding residue and hence corresponds to the $\omega$ angle of the residue to be added ($\omega_i$). The peptide bond length is specified relative to the previous residue in the peptide chain.

A geometry object stores the minimum set of bond lengths, bond angles, and dihedral angles required to uniquely position every heavy atom in the residue. Additional bond angles remain that are not used in our code; these bond angles are defined implicitly. The simplest way to determine which bond lengths and angles are defined for a given amino acid is to print the corresponding geometry object. For example, entering the command `print Geometry.geometry("G")` on the python command line produces the following output:

```
>>> print Geometry.geometry("G")
CA_C_N_angle = 116.642992978
CA_C_O_angle = 120.5117
CA_C_length = 1.52
CA_N_length = 1.46
C_N_CA_angle = 121.38221582
C_O_length = 1.23
N_CA_C_O_diangle = 180.0
N_CA_C_angle = 110.8914
omega = 180.0
```

```
peptide_bond = 1.33
phi = -120
psi_im1 = 140
residue_name = G
```

The following code prints out the default geometries for all amino acids:

```
1 for aa in "ACDEFGHIKLMNPQRSTVWY":
2         print Geometry.geometry(aa)
```

We can construct modified geometries simply by assigning new values to the appropriate member variables. For example, the following code constructs a Gly for which some bond lengths and angles deviate slightly from the default values:

```
1 geo = Geometry.geometry("G")
2 geo.phi=-119
3 geo.psi_im1=141
4 geo.omega=179.0
5 geo.peptide_bond=1.3
6 geo.C_O_length=1.20
7 geo.CA_C_O_angle=121.6
8 geo.N_CA_C_O_diangle= 181.0
```

For amino acids whose side chains require specification of rotamer conformations, there are two ways to specify them. First, we can set rotamers by directly assigning the appropriate values to the correct dihedral angles:

```
1 geo = Geometry.geometry("L")
2 geo.N_CA_CB_CG_diangle=-60.0
3 geo.CA_CB_CG_CD1_diangle=-80.2
4 geo.CA_CB_CG_CD2_diangle=181.0
```

Second, we can set all rotamer angles at once, using the member function `inputRotamers()`:

```
1 geo = Geometry.geometry("L")
2 geo.inputRotamers([-60.0, -80.2, 181.1])
```

In this function call, the angles are listed in order of standard biochemical convention, $\chi_1$, $\chi_2$, $\chi_3$, and so on, for however many $\chi$ angles the amino-acid side chain has.

## CONCLUSION

We have developed a Python library to construct model peptides. Our design goals were to make the library simple, lightweight, and easy-to-use. Using our library, one can construct model peptides in only a few lines of Python code, as long as default bond lengths and angles are acceptable. At the same time, all bond-length and bond-angle parameters are user-accessible and can be modified if so desired. We have verified that our library places atoms correctly. As part of this verification effort, we have found that with increasing peptide length it becomes increasingly important to adjust bond angles appropriately to reconstruct biophysically accurate protein structures.

## ACKNOWLEDGEMENTS

We thank Jeffrey Gray for helpful comments on this work.

### Funding

This work was supported by NIH grant R01 GM088344 to COW. The funders had no role in study design, data collection and analysis, decision to publish, or preparation of the manuscript.

### Grant Disclosures

The following grant information was disclosed by the authors:
NIH: R01 GM088344.

### Competing Interests

The authors declare no competing interests.

### Author Contributions

- Matthew Z. Tien and Claus O. Wilke conceived and designed the experiments, performed the experiments, analyzed the data, contributed reagents/materials/analysis tools, wrote the paper.
- Dariya K. Sydykova and Austin G. Meyer conceived and designed the experiments, contributed reagents/materials/analysis tools, wrote the paper.

### Supplemental Information

Supplemental information for this article can be found online at http://dx.doi.org/10.7717/peerj.80.

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
