# Peer review of "PeptideBuilder: A simple Python library to generate model peptides"

_PeerJ, doi:10.7717/peerj.80_

## Round 0.1 · original submission · Minor Revisions

· Academic Editor

Minor Revisions

Please consider carefully the valuable comments and suggestions of both reviewers in your revision.

Reviewer 1 ·

Basic reporting

In figure 1 C. it is not completely clear as to where the green backbone and blue backbone overlap.

Experimental design

** The submission should clearly define the research question, which must be relevant and meaningful. **

The authors are reporting on a python library used to generate specified peptide and protein conformations as a tool for research. They don't clearly state a "research question" exactly therefore, but they do make clear what the point of the library is.

** Methods should be described with sufficient information to be reproducible by another investigator. **

While the authors note that "The generated models are stored as structure objects using the PDB module of Biopython" they do not explicitly state that this is a dependency and must be installed by the user. i.e. in section 4.1 an import error will be raised immediately by the python interpreter when executing.

>>> import Geometry, PepetideBuilder
ImportError: No module named Bio

whether or not the user later attempts to use Biopython's Bio.PDB.PDBIO().
i.e. the Biopython module is not only required for a limited number PeptideBuilders functions.

Furthermore, a novice user may not realise that the module "Bio" is how the Biopython package is imported. This is a very minor point therefore, but I'd just make it slightly clearer for the benefit of first time users by explicitly stating somewhere in the paper. For example in section 4.1 something either in the text or on line 6:

>>> import Bio.PDB # import Biopython's PDB module.

Validity of the findings

** Does it work **

Yes. While an example use case was provided, no complete unit test was made available to run on PdbBuilder. I tried several examples without any errors being raised.

** The conclusions should be appropriately stated, should be connected to the original question investigated, and should be limited to those supported by the results. **

The authors are reporting on a python library they have written not trying to answer a specific research question. Except perhaps that:

"It is not possible to construct an entire protein structure just from backbone dihedral angles and expect the structure to look approximately correct."

The authors go on in table 1 to quantify (for a number of examples) just how accuracy of the backbone angles effects RMSD. However, in the following sentence I believe there may be a need for further clarification as to what is actually meant, or a rethink on how it is phrased, when they state:

"In particular in tight turns and unstructured loops, backbone bond angles play an important role in determining a biophysically correct structure."

This may be true when building the protein model residue-by-residue from a terminus as PeptideBuilder does and then comparing it to the PDB model; but in reality the precise backbone conformation of unstructured loops will not be as important as the authors might be implying here. The proteins overall fold will be held by contacts between secondary structural elements etc. rather than being affected by the precise conformation of the backbone of residues in flexible loops regions and linker regions.

The authors use a single static PDB structural model to asses whether "a biophysically correct structure" has been, as they put it, "determined" (generated using their script). But as the PDB model is less representative of all the possible backbone bond angles of those loop regions then perhaps other backbone conformations also yield biophysically correct structures.

Unstructured loops will have higher B-factors exactly because they can occupy a variety of different conformations. I do not believe that the evidence the authors have presented means that the specific backbone angles of *unstructured* loops given in the PDB are overly important in the true "biophysically correct" population of occupied protein conformations.

What I believe the authors were trying to state was something like: "if you get the loop region wrong and then continue to build from the terminus (which will be shifted compared to the terminus of the loop in the PDB) then this will affect the remained of the peptide/protein which is built after that loop." I just think as it is currently stated it is ambiguous as to what exactly they mean. Therefore this statement may require a little clarification.

Comments for the author

The authors should consider making the library available through PyPI the python package index.

·

Basic reporting

The paper is overall well-written.

Experimental design

-I don't understand why the 'mean angle' (bottom of p. 2) would need to be calculated from the Dunbrack rotamer library. The rotamers in the library already represent the minima in the energy landscape, as induced from statistical analysis of the observed conformations. There should be no need for further averaging. Why not simply take the most common rotamer conformation?

-p. 5 seems to say that phi is defined in a non-standard way, to refer to the preceeding residue? This is very confusing. If that is indeed the case, then the function should be renamed phi_i1 rather than simply phi. Otherwise this is inviting misuse. It would be much better if phi and psi use the conventional definitions within the residue.

-is there a print function for each object? in particular, it would be helpful to see the data after typing 'print geo' or 'print structure'.

-add_AA seems redundant with add_residue -- could the two functionalities be overloaded on the same name? Otherwise it is not clear from the function names which is which.

Validity of the findings

The authors have shared a simple software package that is well conceived and likely to be use to others.

Comments for the author

Some minor suggestions:

-The title might be less generic if the name of the program was added, ie, "PeptideBuilder: A simple ..."

-p. 3 "tended to perform" might be simply "performed"
-p. 4 first reference to Bio.PDB needs a reference.

-p. 5 mentions other convenience functions. A brief table would help identify these.

-figure 1 is somewhat hard to comprehend. Rainbow coloring of one peptide would help guide the eye from N to C terminus. Also, does A show lots of chain breaks? Even D seems to have more than two termini.

-PDB should be defined on first use.

---

## Round 0.2 · accepted · Accept

· Academic Editor

Accept

no further comments, thank you